# The Properties of Activated Carbons Functionalized with an Antibacterial Agent and a New SufA Protease Inhibitor

**DOI:** 10.3390/ma16031263

**Published:** 2023-02-01

**Authors:** Ewa Burchacka, Katarzyna Pstrowska, Michał Bryk, Filip Maciejowski, Marek Kułażyński, Katarzyna Chojnacka

**Affiliations:** 1Department of Chemistry, Wroclaw University of Science and Technology, Wyspiańskiego Str. 27, 50-370 Wroclaw, Poland; 2Innovation and Implementation Company Ekomotor Ltd., 1A Wyścigowa Street, 53-011 Wroclaw, Poland

**Keywords:** adsorption, *S. aureus* elimination, increased antibacterial activity, antibiotics, phosphonates

## Abstract

*S. aureus* is the cause of many diseases, including numerous infections of the skin. One way to help combat skin infections is to use bandages containing activated carbon. Currently, there are no dressings on the market that use the synergistic effect of activated carbon and antibiotics. Thus, in this study, we point out the adsorption level of an antimicrobial substance on three different active carbons of different origins; by examining the inhibition level of the growth of *S. aureus* bacteria, we determined the number of live cells adsorbed on activated carbons depending on the presence of gentamicin in the solution. In addition, we designed and synthesized a new antibacterial substance with a new mechanism of action to act as a bacterial protease inhibitor, as well as determining the antibacterial properties conducted through adsorption. Our results demonstrate that activated carbons with adsorbed antibiotics show better bactericidal properties than activated carbon alone or the antibiotic itself. The use of properly modified activated carbons may have a beneficial effect on the development and functioning of new starting materials for bacteria elimination, e.g., in wound-healing treatments in the future.

## 1. Introduction

The medical and pharmaceutical industries focus on the external use of activated carbon in extracorporeal devices, such as hemoperfusion columns and materials, and on its use as an oral adsorbent and as a material for dressing production [1]. The activated carbon in dressings is incubated with silver, giving the carbon material placed in the dressing bactericidal properties. Additionally, activated carbon helps to remove unpleasant odors that may be caused by infection. Currently, there are wound-healing materials available on the market with antibacterial properties, such as, for example, silver-plated polyamide fabrics (Silverlon^®^), or those using silver adsorbed on active carbon (Vliwaktiv^®^ Ag) [2]. A novel approach is to adsorb a commercially available antibiotic or a new low-molecular-weight antibacterial agent instead of silver onto the carbon material [3,4]. 

Activated carbons are amorphous solids, mainly composed of elemental carbon (<90%) and characterized by a highly developed surface and a large pore volume. These characteristics of activated carbons can be controlled by modifying the activation conditions [5]. The use of active carbons with appropriate chemical and physical properties increases the effectiveness of the molecules that can be introduced into wound dressings in the future. Our previous studies indicated that the meso- and macropores of the active carbons do not seem to affect the bacterial adsorption process [6]. However, the higher level of oxygen content on the surface of activated carbons is significant for the effective adsorption of Gram-positive and Gram-negative bacteria, such as *S. aureus* and *E. coli* [6]. Rivera-Utrilla et al. studied the adsorption of *E. coli* on two commercially available active carbons. Active carbon samples have been characterized in terms of their surface area, pore size distribution, elemental analysis, mineral substances, and the point of zero charge pH [7]. The adsorption capacity of these carbons increased alongside the increase in both their hydrophobicity and the volume of macropores. The number of bacteria adsorbed on demineralized activated carbon in a solution with a pH equal to the isoelectric point of the coal was negligible [7]. In the case of the adsorption of vitamins, the submicron pores in active carbons impeded the sorption of vitamins A, B1, D, and E. In active carbons with an extensive area of mesopores, the highest level of vitamin sorption was observed [6]. In the case of gentamycin, adsorption occurred in mesopores of the active pharmaceutical carbon tested where a sigmoidal isotherm was observed [8,9]. The application of 6.4 µg/mL of antibiotic on 20 mg of tested pharmaceutical active carbon resulted in approximately 60% adsorption between 5 and 24 h of contact time with the sorbent [9]. The antimicrobial activity of gentamycin at concentrations of 3.6 and 2.3 µg/mL when adsorbed on the tested active pharmaceutical carbon inhibited the growth of *S. aureus* bacteria by 72 and 65.5%, respectively [9]. The previous results provided information on the effective application of pharmaceutical-activated carbon with adsorbed gentamycin as compared to the activity of a pure antibiotic. In this research, the aim of expanding our knowledge on the possible commercial use of activated carbons with adsorbed antibiotics and/or new substances with a new mechanism of action was achieved by analyzing the adsorption level of an antimicrobial substance on three different active carbons of different origins, the inhibition level of the growth of *S. aureus* bacteria, and the number of live cells adsorbed on activated carbons depending on the presence of gentamicin in the solution. In addition, we synthesized a new antibacterial SufA protease inhibitor with the determination of its adsorption level on activated carbon and antibacterial properties against *S. aureus* species. SufA is a family of subtilisin-like serine proteases, with a catalytic triad composed of Asp, His, and Ser [10,11]. SufA was shown to cleave the human fibrinogen, and thus it can inhibit the formation of the fibrin network, suggesting that this protease may play an important role in the development and progression of the clinically important strain of bacteria *F. magna*, which is associated with wounds of the skin, chest, and bone in diabetic patients, as well as tissues of obstetrical and gynecological infections [12,13,14,15,16]. Furthermore, it has been shown that SufA can hydrolyze antibacterial peptides such as LL-37 and the chemokine MIG/CXCL [17]. The SufA protease adheres to the keratinocyte surface and facilitates the colonization of bacteria at the site of infection [17]. Furthermore, SufA releases the adhesion factor from the bacterial cell walls to support bacterial aggregation via the protein–protein interaction and the inactivation of antibacterial peptides produced by the host [17]. One group of compounds that irreversibly inhibits the enzymatic activity of serine proteases (such as SufA, elastase, trypsin, subtilisin, etc.) is a derivative of 1-aminoalkylphosphonate (1-AAP) diphenyl esters [18,19,20,21]. The mechanism of inhibition of serine proteases by 1-AAP diphenyl esters occurs when the hydroxyl group of catalytic oxygen of the serine attacks the electrophilic phosphorus atom (Figure 1A). Following the nucleophilic attack, the pentacoordinate intermediateto form an inhibition product with the loss of a single phenoxy group (Figure 1B). Subsequently, the second phenoxy group is lost in in a process called “complex aging” (Figure 1C). This complex resembles the transition state of the tetrahedral intermediate observed during peptide bond hydrolysis with serine proteases [22].

Our recent experiments focused on the design of molecules Cbz-6-AmNphth^P^(OPh)_2_ that effectively inhibit the SufA protease (Figure 2A) [20]. Cbz-6-AmNphth^P^(OPh)_2_ displayed a k_2_/K_i_ value of 1.08 × 10^4^ M^−1^s^−1^ toward SufA [20]. This phosphonic compound prevented the SufA-mediated degradation of human fibrinogen and showed an antimicrobial activity similar to that of gentamycin against *F. magna*, *S. aureus*, and *E. coli*, with MIC values being 1.4, 14, and 2.8 mg/L, respectively [20]. Additionally, Cbz-6-AmNphth^P^(OPh)_2_ had no cytotoxic effect on normal cell lines (BALB/3T3 and HLMEC), and no inhibitory activity toward the human airway trypsin-like protease (HAT) was observed, which might mean it has potential in in vivo applications [20]. In essence, modification of the structure of the inhibitor Cbz-6-AmNphth^P^(OPh)_2_ was performed in this study (Figure 2). According to the substrate specificity of the SufA protease, the optimization of Cbz-6-AmNphth^P^(OPh)_2_ included the synthesis of a peptide derivative by introducing serine into the P2 position and isoleucine into the position P3 (Schechter and Berger nomenclature, Figure 2B) [23].

The finding of an increased level of antibacterial activity of activated carbon alongside an increase in the adsorbed antibiotic and a new antibacterial substance can contribute to expanding the possibilities for antimicrobial therapies, including improving the conditions of infected wounds. Pathogenic bacteria such as *S. aureus* dominate in the early stage of the infectious process and they are the most common causative agents of skin infection; therefore, *S. aureus* was selected as the representative bacteria in this study [24].

## 2. Materials and Methods

### 2.1. Carbon Activation Method

Active carbons were derived from beech wood (AC1), coconut shells (AC2), and hard coal (AC3). The activation procedure of AC1, AC2, and AC3 was preceded by high-temperature processing (600 °C, 10 °C min^−1^ heating rate, 60 min at the final temperature) under constant inert gas flow (argon, 30 L/h). Physical activation processes were carried out with steam as the activation agent, at a temperature of 700 °C up to the 50% loss of organic mass of the material (calculated as daf—dry, ash free).

### 2.2. Characteristics of the Porous Structure of Activated Carbons

The porous structure of the tested samples was determined by using the thermogravimetric apparatus at 25 °C with the isotherms of carbon dioxide adsorption in the pressure range of 0–700 mmHg and the adsorption/desorption isotherms of benzene in the relative pressure range of p/po 0–1. For interpreting the adsorption isotherms, including the carbon dioxide and benzene adsorption isotherms, the DB (Dubinin–Radushkevich) and BET (Brunauer, Emmet, and Teller) theories were used, respectively [25,26]. For the DR equation calculations, the adopted coefficient β affinity was 0.37, and it was assumed that each carbon dioxide molecule at 25 °C lies flat on the surface of the micropores and covers an area of 0.185 nm^2^ [27]. According to the BET calculations, it was assumed that the surface of the benzene molecule at 25 °C occupies an area of 0.41 nm^2^ [26]. Based on the benzene desorption curve in the range of the relative pressure p/po = 0.96–0.175, the mesopore volume (V_MES_) and mesopore size distribution (S_MES_) as a function of width (2–3; 3–5; 5–10; and 10–50 nm) were calculated. Calculations were performed according to the Pierce method [20], taking into account the amendments to the thickness of the adsorbed layer [28]. The calculations assumed that the pores have the shape of a slot. The average diameter of the mesopore (d_MES_) was calculated from the following equation: d_MES_ = 2V_MES_/S_MES_. The micropore volumes available for benzene vapors (V_MIK_) were calculated as the difference between the volume of benzene adsorbed at p/po = 0.96 and a previously calculated volume of mesopores: V_MIK_ = V0.96−V_MES_ (cm^3^ g^−1^). The micropore volumes of widths below 0.4 nm (submicropores, V_SUB_), which are available to smaller molecules (i.e., carbon dioxide) and inaccessible to larger molecules (i.e., benzene), were calculated. The submicropore volumes were calculated as the difference between the micropore volume calculated from the carbon dioxide adsorption curve (V_MIK_ CO_2_) and the micropore volume calculated from the benzene adsorption curve (V_MIK_) at p/po = 0.96. 

### 2.3. Elemental Content Studies of Active Carbons

Scanning electron microscopy (SEM) was used to the elemental content of the active carbons determination. The activated carbons samples (AC1, AC2, and AC3) were observed using a Quanta 250 FEI scanning electron microscope (Thermo Fisher Scientific Inc., USA) operated at 15 kV. The elemental analysis was performed by means of a Tracor Northern energy dispersive X-ray (EDX) spectrometer (Thermo Fisher Scientific Inc., USA) mounted on the Quanta 250 FEI. The EDX detector was equipped with an ultra-thin light-element window to detect elements with atomic numbers >4. The elemental content results are presented as the mean of content from six independent places on active carbon surface. The results are presented as the percentage of element weight on the active carbon surface ± SD.

### 2.4. Adsorption of Antibacterial Agents on Activated Carbons

The tested activated carbon (20 mg) was measured into an Eppendorf tube and sterilized using a UV lamp, and 1 mL of a 0.5 µg/mL gentamycin solution or 100 µg/mL of Cbz-Ile-Ser-6-AmNphth^P^(OPh)_2_ in PBS was added. In the control sample, 1 mL of PBS was added instead of the antibacterial substance. The samples were incubated at room temperature for 1 h with a gentle shaking of 20 rpm. After incubation, the active carbon solution was removed by centrifugation. Then, the activated carbon tubes were covered with parafilm containing a few small holes made with a needle and were left for 24 h in the desiccator. The supernatant was filtered twice using a pipette with a filter tip. To determine gentamicin concentration, 300 µL of the filtrate was collected from each sample and placed in subsequent Eppendorf tubes, and then 100 µL of ninhydrin solution (5 mg/mL in PBS) was added to each sample. The final mixtures were incubated for 5 min at 95 °C and then cooled for 5 min in an ice bath. After this time, 200 µL of each tube was placed on the plate and the absorbance was measured at 400 nm. The amount of non-absorbed gentamycin on AC1, AC2, and AC3 was calculated by using the standard curve (Appendix A). The concentration of Cbz-Ile-Ser-6-AmNphth^P^(OPh)_2_ in the solution after adsorption was analyzed via high-performance liquid chromatography using a Thermo Scientific Ultimate 3000 instrument with a vacuum degasser (Thermo Scientific Inc., USA). The Interchim^®^ C18 HQ-250/P46 reverse-phase column (250 × 4.6 mm, 5 μm) was used for analysis. HPLC-grade solvents were used throughout the analysis: acetonitrile and water with 0.1% formic acid. The sample injection volume was 10 μL, the flow rate was 1 mL/min, and the detection wavelengths were 280 and 254 nm. The retention time under the described conditions was an average of 12.3–12.7 min for Cbz-Ile-Ser-6-AmNpth^P^(OPh)_2_. The amount of non-absorbed Cbz-Ile-Ser-6-AmNphth^P^(OPh)_2_ on AC1 was calculated using the standard curve (Appendix A). The amount of Cbz-Ile-Ser-6-AmNphth^P^(OPh)_2_ adsorbed on AC1 was calculated as the difference between the initial (100 µg/mL) and the final concentrations in the solution. Results are presented as the percentage of adsorbed Cbz-Ile-Ser-6-AmNphth^P^(OPh)_2_.

### 2.5. Bacterial Growth Reduction

The ability of activated carbons (AC1, AC2, and AC3) with and without adsorbed antibacterial substances to adsorb *S. aureus* (ATTC 6638P) bacteria and the antibacterial properties of these substances were assessed in Mueller–Hinton (MH) broth at a temperature of 37 °C. The overnight culture of tested bacteria was diluted 200 times in fresh MH medium and incubated at 37 °C until the OD_600_ reached 0.3. Next, 1 mL of this culture, which was previously diluted to OD_600_ < 0.01, was added to the 20 mg tested samples of activated carbons with and without adsorbed antibacterial substances. To determine the antibacterial properties of gentamycin and Cbz-Ile-Ser-6-AmNphth^P^(OPh)_2_, the 100 μL of antibacterial substances at concentrations ranging from 0 to 250 µg/mL (in the case of Cbz-Ile-Ser-6-AmNpth^P^(OPh)_2_, the 1% (*v/v*) DMSO concentration was maintained continuously) were introduced into Eppendorf tubes. Then, 900 µL of bacterial culture was added to each tube. Samples prepared in this way were used to test the effect of antibacterial substances on the cell culture without the use of a carbon material. All samples were incubated for 24 h at 37 °C and 100 rpm. Then, 200 µL of each tube was transferred to the plate and the absorbance of each sample was tested at 600 nm. In the case of replicates containing activated carbon, the mixture was taken from above the sediment. To determine the bacterial growth reduction level of the tested samples of AC1, AC2, and AC3 with the adsorbed antibacterial substance, the carbon background (the absorbance due to carbon in the medium) was subtracted from both the tested samples of AC1, AC2, and AC3 with adsorbed antibacterial substances (mean) and the controls (AC1, AC2, and AC3 without adsorbed antibacterial substances). Additionally, the results were compared to the absorbance corresponding to the difference between the bacterial culture (without antimicrobial agents) and the sterile medium to determine the percentage of bacterial growth reduction in the culture that was affected by activated carbons alone or activated carbon with adsorbed antibacterial substances. The percentage of bacteria in the samples was calculated based on the formula: 𝑍% = 𝑃𝐻0 100%, where P is the mean absorbance of the samples after subtracting the carbon background or the absorbance of the control after subtracting the carbon background; *H*0 is the absorbance of the culture not treated with antimicrobial agents, minus the background medium; and Z% is the percentage of bacteria in the samples or controls. To determine the antibacterial properties of gentamycin and Cbz-Ile-Ser-6-AmNphth^P^(OPh)_2_, the data were normalized against a control culture (0μM compound). The IC50 values were calculated in GraphPad Prism through nonlinear regression using the equation model: Y = Y_Bottom_ + (Y_Top_−Y_Bottom_)/(1 + 10((Log IC50−X) * (−1.0)), where Y_Bottom_ and Y_Top_ are plateaued units of the Y axis. IC_50_ gives a halfway response between Y_Bottom_ and Y_Top_, and thus measures the potency of a compound in inhibiting bacterial viability and indicates the reduction of bacterial growth by 50%. 

### 2.6. Quantitative Analysis of Live Bacterial Cells Adsorbed on Activated Carbons

To prepare the samples, 20 mg of activated carbons (AC1, AC2, and AC3), which were previously sterilized via UV, were incubated for 1 h at room temperature in the presence of a gentamicin sulfate solution and PBS without the antibiotic (controls), then drained and left in a desiccator for 24 h. Next, 1 mL of the *S. aureus* bacterial culture with an OD_600_ adjusted to 0.01 was added to the samples and incubated for 24 h at 37 °C and 100 rpm. The supernatant culture was removed by centrifugation. In this way, the prepared samples were suspended in 1 mL of the lysis reagent, then shaken for 10 s and allowed to stand for 15 min. After this time, the samples were shaken again and centrifuged for 5 min at 12,000 rpm. As a result, a clear supernatant was obtained. Then, 100 µL of the supernatant was placed on the test plate in a luminometer, 100 µL of detector solution was added, and the measurement was performed immediately. To calibrate the method, a standard curve was made. For its implementation, a culture of *S. aureus* was prepared in LB medium, which was incubated at 37 °C for 24 h. A series of dilutions was carried out from the baseline to the 10^−7^ samples that were made from the cell culture, and each replicate in the series was analyzed with an ATP kit three times (averaged results). Using the grated plate method (with the LB medium), the number of viable cells was determined for each dilution.

### 2.7. Electron Microscopy Studies of Bacterial Sorption on Active Carbon

The sample of activated carbon AC1 with adsorbed *S. aureus* bacteria and gentamycin was prepared according to the method described in Section 2.5. The control sample without gentamycin was used as the control. The 20 mg samples were fixed in glutaraldehyde (2.5% in cacodyl chloride buffer). After consolidation, the material was washed three times in cacodylic buffer (pH = 6.8) and then dehydrated with a series of ethanol (from 30% to 100% in 10% increments). The dehydrated material was air dried. Dried carbon fragments were glued onto microscope tables using a carbon strip. The material was dusted with gold (10 nm), and after being polished, the samples were subjected to examination with the Auriga 60 scanning electron microscope (Zeiss, Germany) at a beam speed of 2 kV. The samples were observed using secondary electron detectors (SE2 and in-lens).

### 2.8. Synthesis Procedure of Cbz-Ile-Ser-6-AmNphth^P^(OPh)_2_

In the first step of the synthesis procedure, the corresponding aldehyde (**6**) was obtained (Figure 3). 2,6-Dicarboxynaphthyl acid dimethyl ester (**1**) underwent basic hydrolysis to give 2,6-dicarboxynaphthyl acid methyl ester (**2**), which was converted into acid chloride to give 6-carbamoyl-2-naphthalenecarboxylic acid methyl ester (**3**). Compound **3** was dehydrated to give 6-cyano-2-naphthalenecarboxylic acid methyl ester (**4**). In product 4, the ester group was reduced to hydroxyl using lithium borohydride. The resulting 6-cyano-2-hydroxymethylnaphthalene (**5**) was oxidized using the Swern method, which converted the hydroxyl group to an aldehyde (**6**). Compound 6 was converted to benzyloxycarbonylamino-[(6-cyano) naphthyl] methanophosphonic acid diphenyl ester (**7**) through the amidoalkylation reaction. In the next step, the imino ether was obtained under anhydrous conditions, and was then transformed into an amidine group using ammonium in methanol, producing compound **8** (Figure 3g). In the next step, the Cbz-protective group was removed via the hydrogenolysis reaction over Pd on carbon, and crud product **9** was directly used for the coupling of Cbz-Ile-Ser(t-Bu)-OH dipeptide with HOBt and EDC as reagents in a 30% TFE/dichloromethane solution. In the last step, the desired compound **11** was obtained through the deprotection of compound **10** using a 50% solution of TFA in dichloromethane. Mass spectroscopy and nuclear magnetic resonance were used to confirm the presence of Cbz-Ile-Ser-6-AmNphth^P^(OPh)_2_ (Appendix A).

#### 2.8.1. 6-(Methoxycarbonyl)-2-naphthoic Acid (**2**)

The dimethyl naphthalene-2,6-dicarboxylate (**1**, 0.02 mol) was heated at 80–90 °C in 1,4-dioxane (30 mL) until completely dissolved. Next, the solution of KOH in MeOH (1.32g KOH/10 mL MeOH) was added dropwise and the mixture was heated for 2h at 90°C. The reaction mixture was cooled to room temperature and the crystalline product was filtered, washed with diethyl ether, dried, and dissolved in water (250 mL). The insoluble material was removed by filtration. The clear filtrate was acidified with 2M HCl to pH 3. The precipitate was filtered and dried under a vacuum over P_2_O_5_. Yield: 83%; mp: 275 °C. ^1^H NMR (300 MHz, DMSO-d6, ppm): δ 3.96 (s, CH_3_, 3H), 8.01–8.06 (m, Ar, 2H), 8.19–8.23 (m, Ar, 2H), 8.66 (s, Ar, 2H), 8.7 (s, OH, 1H); *m/z* [M+H]+ calc. for C_13_H_10_O_4_: 231.0657; found: 231.0737 (Appendix A).

#### 2.8.2. Methyl 6-carbamoyl-2-naphthoate (**3**)

A solution of 6-(methoxycarbonyl)-2-naphthoic acid (**2**, 0.016 mol) in ethylene chloride (120 mL) was mixed with thionyl chloride (60 mL) and the reaction mixture was heated at 75 °C for 3 h. The volatile components were removed under reduced pressure, redissolved in dry toluene, and evaporated again. The oily residue was dissolved in anhydrous methylene chloride (40 mL) and a 7M ammonia solution in methanol (5 mL) was added. The reaction was carried out for 1h at room temperature. The product that precipitated as a white solid was filtrated and dried in P_2_O_5_. Yield: 93%. ^1^H NMR (300 MHz, DMSO-d6, ppm): δ 3.92 (s, CH_3_, 3H), 8.02 (d, *J* = 6.7 Hz, Ar, 2H), 8.23 (d, *J* = 10.7 Hz, Ar, 2H), 8.69 (d, *J* = 11.5 Hz, Ar, 2H), 13.29 (s, NH_,_ 1H),; *m/z* [M+H]+ calc. for C_13_H_11_NO_3_: 230.0817; found: 230.0812 (Appendix A).

#### 2.8.3. Methyl 6-cyano-2-naphthoate (**4**)

Methyl 6-carbamoyl-2-naphthoate (**3**, 0.01 mol) was suspended in 1,4-dioxane (35 mL), and pyridine (2.4 mL) was added to it. The solution was cooled to 0 °C in an ice bath and trifluoroacetic anhydride (20 mL) was added dropwise. The reaction mixture was allowed to warm to room temperature and was continued for 48 h. The reaction mixture was poured into water (300 mL) and extracted with ethyl acetate (3 × 70 mL). The combined organic extracts were washed with water (4 × 100 mL) and dried in MgSO_4_. After filtration, the volatile components were removed under vacuum, and the resulting oil was dissolved in chloroform, passed through a pad of silica gel, and evaporated to dryness, yielding the final product as a white solid. Yield: 44%; mp: 147–150 °C. ^1^H NMR (300 MHz, DMSO-d6, ppm): δ 3.93 (s, CH_3_, 3H), 7.87 (dd, *J* = 1.6/8.5 Hz, Ar, 1H), 8.08–8.18 (dd, *J* = 37.0 Hz, Ar, 2H), 8.33 (d, *J* = 8.7 Hz, Ar, 1H), 8.65 (s, Ar, 1H), 8.72 (s, Ar, 1H); *m/z* [M+Na]+ calc. for C_13_H_9_NO_2_Na: 234.0531; found: 234.0593 (Appendix A).

#### 2.8.4. 6-(Hydroxymethyl)-2-naphthonitrile (**5**)

LiBH_4_ (0.09 mol) was added to the solution of 6-cyano-2-naphthoate (**4**, 0.008 mol) in THF (100 mL). The mixture was stirred for 30 min at room temperature, which was followed by the addition of ethyl alcohol (100 mL). The reaction was carried out at room temperature for 24 h. Then, a 5% aqueous citric acid solution was added dropwise and volatile components were removed under reduced pressure. The aqueous phase was extracted with chloroform (3 × 100 mL) and combined organic fractions were washed with water (2 × 100 mL) and brine (2 × 100 mL), which were then dried over MgSO_4_. The solvent was evaporated under reduced pressure, yielding the desired alcohol. Yield: 96%; mp: 110–113 °C. ^1^H NMR (300 MHz, CDCl_3_, ppm): δ 2.02 (s, OH, 1H), 4.91 (s, CH_2_, 2H), 7.60 (dd, *J* = 1.3/8.5 Hz, Ar, 2H), 7.87–7.91 (m, Ar, 3H), 8.20 (s, Ar, 1H); *m/z* [M+Na]+ calc. for C_12_H_9_NONa: 206.0582; found: 206.0532 (Appendix A).

#### 2.8.5. 6-Formyl-2-naphthonitrile (**6**)

Oxalyl chloride (0.1 mol) was dissolved in freshly dried dichloromethane (40 mL), and the solution was cooled to −60 °C, which was followed by a dropwise addition of dimethyl sulfoxide (0.27 mol) dissolved in 40 mL of dry dichloromethane. After 15 min, 6-(hydroxymethyl)-2-naphthonitrile (**5**, 0.06 mol) dissolved in 100 mL of dry dichloromethane was added dropwise. The reaction was carried out at −60 °C for 2 h. The reaction was quenched with triethylamine (2 mL), which was followed by the addition of a saturated ammonium chloride solution (80 mL). The mixture was extracted with methylene chloride and the combined organic fractions were washed with brine (50 mL) and dried over MgSO_4_. The solution was filtered and evaporated to dryness, yielding crude aldehyde which was used directly in the amidoalkylation reaction.

#### 2.8.6. Cbz-6-CN-Nphth^P^(OPh)_2_ (**7**)

A mixture of 6-formyl-2-naphthonitrile (**6**, 0.011 mol), benzyl carbamate (0.011 mol), and triaryl phosphate (0.01 mol) was dissolved in 50 mL of glacial acetic acid. The solution was stirred at 80–90 °C for 3 h. The volatile components were removed under vacuum and the product was crystallized from methanol at −20 °C. It was prepared from 6-formyl-2-naphthonitrile, benzyl carbamate, and triphenyl phosphate via an amidoalkylation reaction in acetic acid. The product was obtained as a white solid. Yield: 25%; mp: 175°C. ^31^P NMR (243 MHz, CDCl_3_, ppm): δ 12.85 (s); ^1^H NMR (300 MHz, CDCl_3_, ppm): δ 5.07* (d, *J* = 12.1 Hz, CH_2_, 1H), 5.15* (d, *J* = 12.1 Hz, CH_2_, 1H), 5.76 (dd, *J* = 9.1/22.8 Hz, CHP, 1H), 6.19 (m, NH, 1H), 6.87–7.37 (m, Ar, 15H,), 7.59 (dd, *J* = 1.0/8.5 Hz, Ar, 1H), 7.72–7.88 (m, Ar, 4H), 8.19 (s, Ar, 1H); *m/z* [M+Na]+ calc. for C_32_H_25_N_2_O_5_PNa: 571.1399; found: 571.1322 (Appendix A).

#### 2.8.7. Cbz-6-AmNphth^P^(OPh)_2_ (**8**)

Cbz-6-CN-Nphth^P^(OPh)_2_ (**7**, 1 mmol) was dissolved in a mixture of anhydrous ethyl alcohol (1.6 mL) and freshly dried chloroform (15 mL). Next, the solution was saturated with gaseous HCl and the reaction was performed at −20 °C for 4 days. The product was precipitated with diethyl ether, filtered, and dried in a vacuum over P_2_O_5_. The intermediate imino ether obtained was dissolved in dry methanol (50 mL) and a 7N ammonia solution in methanol (0.15 mL) was added. The reaction was performed at room temperature for 1 h. Subsequently, the volatile components of the reaction mixture were removed in a vacuum. The resulting oil was redissolved in fresh methanol and refluxed for 8 h. The reaction mixture was evaporated, and the oily product was further purified in silica gel using CHCl_3_:MeOH: AcOH/90:10:1 (*v/v/v*) as the eluent, yielding a white solid final product. Yield: 30%. ^31^P NMR (243MHz, d-MeOH, ppm): δ 12.98 (s); 1H NMR (400 MHz, CDCl_3_, ppm): δ 5.06* (d, *J* = 12.5 Hz, CH_2_, 1H), 5.15* (d, *J* = 12.3 Hz, CH_2_, 1H), 5.83 (dd, *J* = 22.8 Hz, CHP, 1H), 6.97-7.35 (m, 15H), 7.89 (dd, *J* = 8.4/22.1 Hz, 2H), 8.08 (t, *J* = 7.8 Hz, 2H), 8.27 (s, 1H), 8.44 (s, 1H); *m/z* [M+H]+ calc. for C_32_H_29_N_3_O_5_P: 566.1839; found: 566.1829 (Appendix A).

#### 2.8.8. Cbz-Ile-Ser(t-Bu)-OH

Fmoc-Ser(t-Bu)-OH (1.5 equiv) was coupled with the Rink amide resin in CH_2_Cl_2_ (loading: 0.32 mmol/g) using HBTU (1.50 equiv) and DIPEA (3.00 equiv) for 4 h. The resin was washed with CH_2_Cl_2_ and methanol was added. After 20 min, the methanol was removed and the resin was washed with DMF. For the deprotection of the Fmoc group, a 20% piperidine solution in DMF was used and the coupling of Cbz-Ile-OH (1.5 equiv) was performed using HBTU (1.50 equiv) and DIPEA (3.00 equiv). The resin was washed with DMF, Hex, and CH_2_Cl_2_ (3 × 15 mL). The peptide was cleaved with 3a 0% TFE solution in CH_2_Cl_2_ (3 × 10 mL, 30 min). The volatile components of the mixture were removed by evaporation and the residues were precipitated and washed with Et_2_O (20 mL) to afford the crude dipeptide that was purified using the HPLC method (retention time of 16.92 min). *m/z* calc. for C_21_H_32_N_2_O_6_: 408.2; [M+H]+ found: 409.4.

#### 2.8.9. NH_2_-6-AmNphth^P^(OPh)_2_ (**9**)

In a 50 mL two-necked flask equipped with a magnetic stirrer and bubble, 0.035 mmol of Cbz-6-AmNphth^P^(OPh)_2_ (**8**) was placed and dissolved in 10 mL of methanol; then, a small amount of Pd-C was added. Hydrogen gas was introduced into the reaction mixture for about 2.5 h. The reaction mixture was filtered through a syringe filter. The solvent of the filtrate was evaporated on a rotary evaporator. The 9.3 µmol of NH_2_-6-AmNphth^P^(OPh)_2 2_ was obtained, which was immediately used in the coupling reaction with the dipeptide.

#### 2.8.10. Cbz-Ile-Ser-6-AmNphth^P^(OPh)_2_ (**11**)

The mixture of Cbz-Ile-Ser(*t*-Bu)-OH dipeptide (9.3 µmol), HOBt (9.3 µmol), and EDC (9.3 µmol) was dissolved in a 30% TFE/CH_2_Cl_2_ solution and stirred for 5 min. NH_2_-6-AmNphth^P^(OPh)_2_ (**9**, 9.3 µmol) was added to the mixture. The reaction was carried out at room temperature for 24 h. At this time, an additional amount of HOBt (18.6 µmol) and EDC (18.6 µmol) was added, and the reaction was continued for 24 h. The volatile components of the mixture were removed by evaporation and a 50% TFA solution in CH_2_Cl_2_ was added. After 2 h of the deblocking reaction, the volatile components of the mixture were removed by evaporation. The desired product was purified using the HPLC method. Yield: 30%. ^31^P NMR (121 MHz, d-MeOH, ppm): δ 13.67 (s), 13.64 (s); ^1^H NMR (600 MHz, d-MeOH, ppm): δ 8.45–6.69 (m, Ar-H, 22H), 6.23–6.00 (m, CHP, 1H), 5.34–4.79 (m, CH_2_, 2H), 4.65–4.52 (m, OH, 1H), 3.92 (dd, *J* = 25.9/6.9 Hz, CH, 1H), 3.79–3.72 (m, CH, 1H), 1.80–1.58 (m, CH, 1H), 1.30–1.04 (m, CH_2_, 2H), 0.92–0.55 (m, 2×CH_3_, 6H); *m/z* calc. for C_41_H_44_N_5_O_8_P: 765.3; [M+H]+ found: 766.7 (Appendix A).

### 2.9. SufA Inhibition Studies

The assay buffer was 0.1M Tris-HCl (pH 8.0) supplemented with 0.01M KCl. The substrate used was ABZ-Ile-Ser-Lys-ANB (ABZ: aminobenzoic acid, ANB:5-amino-2-nitrobenzoic acid). The enzyme inhibition assay was performed using 96-well microtiter plates (COSTAR, Corning Inc., Gdansk, Poland) at 37 °C with an excitation wavelength of 320 nm for ABZ and emission wavelength of 410 nm. The SufA protease was incubated for 10 min with the tested compound (inhibitor final concentration ranging from 0 µg/mL to 38.25 µg/mL), and then the measurement was taken for 5 min. The standard deviation for the values presented is the mean of two independent experiments. All measurements were made using a SpectraMax Gemini XPS spectrofluorometer (Molecular Devices, Sunnyvale, CA, USA). The percentage of inhibitory activity was measured using the incubation method and was calculated using the equation: %inhibition = [*RFU*_test_/*RFU*_(-)_]*100%, where *RFU* represents the relative fluorescence units, *test* refers to the signal from reactions containing an inhibitor, and (-) represents the signal from the control reactions (without an inhibitor).

## 3. Results

### 3.1. AC Characteristics

The properties of activated carbons that determine their final application include their pore size distribution (mainly micropores) and their high surface area (1000–3000 m^2^/g). In this study, carbons were derived from beech wood (AC1), coconut shells (AC2), and hard coal (AC3), and the carbonization process was used for their activation. In this step, three of the carbons tested (precursor materials) reacted with the activating agent, generating materials with a high porosity that were used as activated carbons (AC1, AC2, and AC3). In AC2 and AC3, the water vapor agent was additionally used in its physical activation. AC1 contains 4.4% dry ash, whereas AC2 contains an amount that is two times lower, at 2.2% (Table 1). The highest dry-ash-free coefficient was 5.3%, and the lowest humidity was 0.7%, which was determined for AC3 (Table 1). The humidity for AC1 and AC2 was six times higher and more than four times higher compared to AC3, respectively (Table 1). The smallest surface of micropores and mesopores (515 m^2^/g) was observed for AC1 (Table 1). Analysis of the porous structure indicated that both AC2 and AC3, for the most part, consist of micropores with widths from 0.4 to 2 nm (data not shown). The micropores occupied a surface measuring 1107 and 946 m^2^/g in AC2 and AC3, respectively. The micropore area in AC1 was about half the size of AC2-3 and equaled 472 m^2^/g. The area of submicropores with widths of <0.4 nm was 515 m^2^/g for AC1, whereas submicropores were not observed for AC2 and AC3 (Table 1). The mesopore area of AC3 was 61.6 m^2^/g, making this sample the most abundant in mesopores as compared to the other samples tested (AC1 and AC2, Table 1). The area of mesopores in AC1 and AC2 was approximately 43 and 44 m^2^/g, respectively. 

The elemental analysis of active carbons AC1, AC2, and AC3 indicate that the AC1 revealed oxygen contents of 14.0% (Table 2). The content of oxygen groups on the surface of AC1 was more than two times higher than that of active carbons AC2 (6.1 ± 1.5%). The oxygen contents on the AC3 surface was 11.3 ± 1.4 (Table 2).

### 3.2. Inhibition Studies of Cbz-Ile-Ser-6-AmNphth^P^(OPh)_2_

The inhibitory properties of Cbz-Ile-Ser-6-AmNphth^P^(OPh)_2_ (**11**) against the SufA protease are presented in Figure 1. The data obtained show that Cbz-Ile-Ser-6-AmNphth^P^(OPh)_2_ inhibited SufA by 39.4% ± 3.36 at a concentration of 38.25 µg/mL. A similar percentage of the enzyme inhibition value was observed for the 30.6 µg/mL concentration of compound **11** (36.2% ± 1.42). Halving the concentration of compound **11** to 15,3 µg/mL reduced the activity of the SufA protease by only 25.4% ± 3.4. At the lowest concentration of inhibitor **11**, which was 7,65 µg/mL, the inhibition of enzyme activity was negligible (4.35% ± 3.0).

### 3.3. Adsorption Studies

The results presented in Table 3 suggest that the adsorption of gentamycin (Gt) on the activated carbons AC1, AC2, and AC3 occurred. The degree of adsorption is high (just below 100%), which means that the vast majority of the gentamicin sulfate used was adsorbed on the surface of all three active carbons (AC1, AC2, and AC3). However, the highest value of gentamycin adsorption, 98.33%, occurred on the active carbon from coconut shells (AC2) and was observed after 1 h of contact time. The adsorption of Cbz-Ile-Ser-6-AmNphth^P^(OPh)_2_ on AC1 was noted to be at the level of 100% after 1 h of contact (Table 3). 

### 3.4. Bacterial Growth Reduction

The graph of the reduction in the dependence of *S. aureus* bacterial growth of on the concentration of the inhibitor Cbz-Ile-Ser-6-AmNphth^P^(OPh)_2_ (**11**) is presented in Figure 2. Compound **11** showed an inhibition of around 50% of *S. aureus* growth at a concentration of 250 µg/mL (Figure 2A: red, dashed line). The experimentally determined MIC and IC_50_ values for the commercially available antibiotic gentamicin were 1.0 and 0.326 µg/mL toward *S. aureus*. A concentration of 100 µg/mL of Cbz-Ile-Ser-6-AmNphth^P^(OPh)_2_, which corresponded to the concentration of **11** adsorbed on AC1, was used as the control in the bacterial growth reduction studies (Figure 3A: blue, dashed line). Cbz-Ile-Ser-6-AmNphth^P^(OPh)_2_ (100 µg/mL) retained a 31% growth of *S. aureus* (Figure 2A, Figure 3A: blue, dashed line). The tested material—AC1 with adsorbed Cbz-Ile-Ser-6-AmNphth^P^(OPh)_2_—reduced *S. aureus* by 55% (Figure 3A: green, striped column) compared to the control sample. Additionally, the adsorption level of *S. aureus* on AC1 (without antibacterial agents) was examined and was found to be 40% (Figure 3A: green column). 

The study of the effect of gentamicin on activated carbon showed that the highest activity among the four antibiotic–carbon systems was in carbon AC1, which reduced the number of bacterial cells in the culture to 77.3% compared to the initial culture without being affected by the activated carbon, antibiotic, or the synergistic effect of both of these factors (Figure 3A: green, dotted column). The lowest activity of the tested samples was observed in AC3, where the decrease in the number of cells in the culture was 63.4% (Figure 3C: blue, dotted column). Additionally, it should be noted that AC2 with adsorbed gentamycin revealed similar results as compared to AC1 with adsorbed gentamicin, as both had a bacteria content in the culture of 22% (Figure 3B: grey, dotted column; Figure 3A: green, doted column). The most desirable effect of activated carbon alone without an antibiotic on the number of bacteria was also shown by the AC1 because it caused the number of bacterial cells in the culture to be reduced to 38.1% of the initial culture (Figure 3A: green column). The lowest reduction in *S. aureus* cells in culture, ~5%, was observed for AC2 (Figure 3B: grey column). Although AC2 showed the lowest level of bacterial reduction of approximately 5%, the adsorption of the antibiotic (Gt) on this material improved bacteria reduction in the culture by as much as 72.3% (Figure 3B: grey, dotted column. The antibiotic concentration of 0.492 µg/mL, corresponding to the adsorbed concentration of AC2, reduced the number of bacteria in the culture by 45.7% (Figure 3B: red dashed line). In the case of AC3, there was a high level of adsorbed gentamycin (0.449 µg/mL) and a fairly high degree of bacterial adsorption (29.5%) by the AC3 material itself (Figure 3C: blue column). The degree of the number of bacteria in the culture with active carbon derived from hard coal (AC3). amounted only to 36% despite the use of AC3 with an adsorbed gentamycin (Figure 3C: blue, dotted column). 

### 3.5. Quantitative Analysis of Live Bacterial Cells

The gentamycin adsorbed on all the tested activated carbons (AC1, AC2 and AC3) exhibits antibacterial activity toward bacterial cells adsorbed on carbon material. This is evidenced by the results presented in Table 4, as the number of live bacteria cells adsorbed on activated carbon incubated with the antibiotic was 23 × 10^7^, 2.2 × 10^7^, and 1.7 × 10^7^ CFU/mL toward AC1, AC2, and AC3, respectively. A higher number of viable bacterial cells adsorbed on activated carbon that did not have contact with the antibacterial substance was shown for AC1, where the calculated number of live *S. aureus* cells was 41 × 10^7^. The results presented demonstrate that gentamycin adsorbed in all tests of AC reduced the number of viable cells by 56, 61, and 60 % in the AC1, AC2, and AC3 materials, respectively. 

### 3.6. Electron Microscopy Studies of Bacterial Sorption on Active Carbon

Clearly, the electron microscopy studies (EMS) indicated that *S. aureus* adhered in a greater amount to the surface of AC1 when gentamicin was not adsorbed to the activated carbon (Figure 4). The application of AC1 with adsorbed gentamicin led to a reduced number of bacteria adhering to the carbon surface, which is a result of the antibacterial properties of gentamicin (Figure 4). However, the EMS studies reveal that despite the inhibition of bacterial growth through gentamycin action, the AC1 sorption capacity toward bacteria is still preserved, which explains the increased level of bacterial reduction relative to the action of the pure antibiotic in the composite material of AC1 with an adsorbed antibiotic.

## 4. Discussion

The active carbon AC2 derived from coconut shells was selected as the most effective sample for the gentamycin adsorption process (98.3% of antibiotic adsorption). However, the level of the antibiotic adsorption of all tested materials was not significantly different, and the adsorption values were high regardless of the carbon used, being in the range of 89.5–98.3%. AC1 and AC3 revealed a level of *S. aureus* adsorption of 38.1% and 29.5%, respectively (Figure 3A: green column; Figure 3C: blue column). Studies on the use of active carbons from coconut shells (AC2) as a bacterial sorbent resulted in the adsorption of *S. aureus* at a low level of 4.6% (Figure 3B: grey column). Our previous studies indicated that the adsorption of *S. aureus* was 50%, 90%, 80%, and 90% (6 h of cells being in contact with the adsorbent) for a pharmaceutical active carbon, AC1, AC2, and AC3, respectively [6]. In this case, the sorption capacity of carbons was much higher due to the amount of sorbent used being seven times greater and the bacteria content being lower. However, we also observed that the carbon surface oxygen group content was dependent on the adsorption of bacterial cells. The level of oxygen content on the surface of AC1, AC2, and AC3 was 14, 6, and 11% (Table 2), which is confirmed by the fact that the higher the content of oxygen groups, the better the sorption of the *S. aureus* bacteria. From the obtained results, no relationship between the size of pores in activated carbons and their properties of bacterial adsorption was observed. It seems, therefore, that the meso- and macropores of activated carbons may not affect bacterial adsorption. Although AC2 revealed the highest level of adsorbed gentamycin, it also revealed the lowest level of bacteria sorption among all tested samples, and the reduction in bacteria in the culture with the application of AC2 with adsorbed antibiotics was high and equal to 77% (Figure 3B: grey, dotted column). Furthermore, the antibiotic solution with a concentration of 0.492 µg/mL (which corresponds to the concentration of gentamycin adsorbed on AC2 at 1h) was introduced into the culture without activated carbon, and it reduced the number of bacterial cells by 32% less than AC2 with the adsorbed antibiotic. The reduction in *S. aureus* increased by 35.3% using AC1 with adsorbed gentamycin as compared to the pure antibiotic at a concentration of 0.448 µg/mL (Figure 3A: green, doted column; red dashed line). Therefore, AC1 derived from beech was selected as the most effective material for reducing bacteria, as it was equal to 77.3%. Although AC3 showed the lowest increase in bacterial reduction of 20.9%, as compared to the concentration of the pure antibiotic corresponding to the concentration adsorbed on this material, the results obtained demonstrated that all of the tested activated carbons support the antibacterial agent in increasing the reduction in bacterial growth (Figure 3). Furthermore, our EMC studies revealed that despite the antibacterial properties of gentamycin adsorbed on AC1, the sorption capacity of *S. aureus* at a limited level still occurs, which explains the increased level in bacterial reduction relative to the action of pure antibiotics. Moreover, this part of the study shows that gentamicin sulfate adsorbed on activated carbon causes the death of *S. aureus* cells, which was additionally confirmed by a quantitative analysis of viable bacterial cells. The number of viable cells was reduced by 56, 61, and 60% with AC1, AC2, and AC3, respectively. This relationship was consistent with the amount of adsorbed antibiotic, the highest level of which was 98.33% for AC2. 

In this study, the synthesis of the new compound Cbz-Ile-Ser-6-AmNphth^P^(OPh)_2_ (11) was also presented. Previous studies showed the inhibitory activity of Cbz-6-AmNphth^P^(OPh)_2_ against the SufA protease, displaying a k_2_/K_i_ value of 10 800 M^−1^s^−1^ [20]. In addition, Cbz-6-AmNphth^P^(OPh)_2_ prevented SufA-mediated human fibrinogen hydrolysis in vitro and exhibited potent antibacterial activity against *F. magna*, *S. aureus*, and *E. coli*; thus, it may lead to the development of new antibacterial drugs with a new mechanism of action in the future [20]. The specificity of the SufA protease substrate was used for expanding the research on the SAR (structure–activity relationship) of compound Cbz-6-AmNphth^P^(OPh)_2_. The preferred substrate sequence of the SufA protease is Ile-Ser-Lys/Arg (P3–P2–P1; Schechter nomenclature) [29]. Based on the substrate recognition pattern, we introduced a procedure for the synthesis of Cbz-Ile-Ser-6-AmNphth^P^(OPh)_2_ (**11**) to determine its influence on the inhibitory activity toward SufA and its antibacterial properties when adsorption was carried out. Surprisingly, compound **11** at a concentration of 38,25 µg/mL revealed an SufA inhibition of only 39.4% ± 3.36, whereas the inhibitory activity of Cbz-6-AmNphth^P^(OPh)_2_ was almost two times higher at the same concentration (data not shown). Data from the literature demonstrated that the peptidyl derivatives of 1-AAP increase the inhibitory activity, selectivity, and specificity of action against serine proteases. As an example, the tripeptidyl derivative of Boc-Val-Pro-Phe^P^(OC_6_H_4_-4-SCH_3_)_2_, which corresponds to the amino acid sequence of the subtilisin substrate, exhibits an inhibitory activity that is approximately 40 times higher (k_2_/K_i_ = 114 380 ± 4 570 M^−1^s^−1^) than the single amino acid analog of Cbz-Phe^P^(OC_6_H_4_-4-SCH_3_)_2_, with a value of k_2_/K_i_ being 2 802 ± 144 M^−1^s^−1^ against subtilisin [18]. The antibacterial activity of compound **11** was significantly lower than the activity of non-peptide derivatives of Cbz-6-AmNphth^P^(OPh)_2_. Our previous studies indicated that the MIC value of Cbz-6-AmNphth^P^(OPh)_2_ against *S. aures* was 14 µg/mL, whereas Cbz-Ile-Ser-6-AmNphth^P^(OPh)_2_, at a concentration of 250 µg/mL, inhibited only 50% of *S. aureus* growth. These data suggest that the introduction of Ile and Ser residues at the P2 and P3 positions is not optimal for binding the inhibitor to the SufA active site. It should be noted that the enzyme shows substrate specificity for Arg and Lys residues in the P1 position, and peptide derivatives of Arg^P^(OPh)_2_ and/or Lys^P^(OPh)_2_ can significantly improve the inhibitory properties of the SufA protease. Our research will continue in this direction. Additionally, the **11**-Cbz derivative of 1-AAP can bind differently to the SufA protease, and additional amino acid residues in P2 and P3 hinder the ability to covalently bind to the serine residue at the SufA active site through steric hindrance. Thus, crystallographic and/or molecular modeling studies in this direction will require more attention in the future. Despite the lack of improved antibacterial and inhibitory properties against SufA of **11** as compared to Cbz-6-AmNphth^P^(OPh)_2,_ the synthesized compound **11** was 100% absorbed into the tested carbon AC1 at a 100 µg/mL. Moreover, Cbz-Ile-Ser-6-AmNphth^P^(OPh)_2_ (100 µg/mL) retained the growth of *S. aureus* at 31% (Figure 2A). The tested material—AC1 with adsorbed Cbz-Ile-Ser-6-AmNphth^P^(OPh)_2_—increased the reduction in *S. aureus* and an 86% bacteria reduction was observed (Figure 3A: green, striped column). 

## 5. Conclusions

Studies conducted in vitro prove that the tested antibiotic was adsorbed on activated carbons while maintaining bactericidal activity. The adsorption properties of the antibiotic remained at the level of approximately 90%. The results obtained indicated a synergistic effect between the antibiotic and activated carbon. These charcoal–antibiotic systems kill far more bacterial cells in culture than either antibiotic or activated charcoal alone. The observed reduction in the number of bacterial cells of all tested activated carbons with antibiotics was between 63.3 and 77.3%. In addition, the experiment showed the range of the adsorption properties of activated carbon; e.g., there was adsorption of organic molecules, as well as live bacterial cells. Additionally, the adsorption of the new compound Cbz-Ile-Ser-6-AmNphth^P^(OPh)_2_ (**11**) as an inhibitor of the serine protease SufA also indicated that the activated carbon antibacterial substance material improved bacterial growth reduction in vitro. Further analogous studies can be performed for activated carbons of even more diverse origins, as well as to test the effect of another antibiotic or use different microorganisms to determine the effect of such a system on cell culture. The presented experiment and its results taking into account future cell cytotoxicity research may be the basis for the introduction of activated carbon with adsorbed antibiotics in the use of plasters, bandages, and sterile dressings in general, especially since the experiment was carried out at a pH of 7.4, i.e., one that resembles the pH of human blood. Such materials would demonstrate antibacterial properties and could eliminate the unpleasant odor emanating from the wound. These dressings would have properties similar to those already introduced on the market that contain activated carbon with adsorbed silver ions [4].

## Data Availability

Not applicable.

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
