# Peer review of "The Properties of Activated Carbons Functionalized with an Antibacterial Agent and a New SufA Protease Inhibitor"

_materials, 2023, doi:10.3390/ma16031263_

Round 1

Reviewer 1 Report

The manuscript of Burchacka et al. contains much functional information, but the present form, including the Supporting Materials, suffers from some shortcomings. The current version needs a deep revision.

The referee's concerns are summarized below.

- The authors need to explain what additional information holds the expressions active carbon and SufA protease inhibitor in the Keywords section. They are already in the title.

- In line 38, the "highly developed internal surface" needs some clearance. What did the authors want to express? If there are internal surfaces, there should be outer surfaces, too. How can we differentiate them?

Line 47 further complicates the situation because the authors missed mentioning which surface area characterizes the active carbons, inner, outer, or both.

- in line 51, activated seems more appropriate.

- In line 120, what does carbonization mean in the case of AC3? Since AC1 and AC2 come from organic sources, "carbonization" is correct, but AC3 is predominantly carbon, and the term, therefore, makes no sense.

- In line 133, BET and not BMT?

- The antibiotics discussed should have a detailed scheme in the Introduction or at the beginning of the Materials and Methods. The 'detailed' means no abbreviation for amino acids.

- In line 158, secured means sealed or covered? Please clarify it.

- Cbz and Npth need resolution. Additionally, the Nphth might be better.

- The recovery experiment description of adsorbed antibiotics from the active carbons is missing. Although the reader can gather this information later, the authors should devote a few separate sentences to the desorption experiment. Later, in a table, the authors mention the antibiotic content of the carbons, so they have done this experiment.

On the other hand, the 2.4.1 and 2.4.2 sections contain repetitions. Because the technology is only minimally different, they must combine these two sections.

- Generally, in Section 2.6, the authors describe the syntheses of all intermediates of Cbz-Ile-Ser-6-AmNpthP(OPh)2. Unfortunately, they included the NMR spectra of only the final product in the SI file (but its carbon is missing). Because in the experimental section, NMR data are only in extracted form, and without the purity, the reader misses some information about the authors' procedures, they should put spectra of all prepared compounds into the SI file. Seeing is believing.

Based on the yield in section 2.6.1, we can also assume the formation of a doubly hydrolyzed product. The lack of purification and excess of KOH also suggest this byproduct. The NMR spectrum can confirm or reject this assumption.

In lines 256-257, filtered is more appropriate than filtrated.

In line 258, what does "product participated with 2M HCl" mean?

- Table 1 needs a redesign.

The meaning of "Proximate analysis" is unclear. Humidity and Ash experiments are not "proximate" but require direct experiments.

Column 2 is confusing. Additionally, what value is 4.4.0? 4.40 or 44.0? The table has a footnote, but it is unclear which results show "dry and ash-free" value(s).

- Figure 1 seems a better location in the Materials and Methods section.

- ANB (occurs in lines 370, 442, 445) needs a resolution.

- Table 2 is confusing. Is the "concentration adsorbed on AC" microgram/milliliter indeed? Not microgram/milligram? If ug/mL is correct, please explain the reason. Usually, the characterization of solid materials uses weight and not volume.

Please also explain the difference of values for compound 11 in column 2. What is the difference between 0 ug/mL and 'not determined' (or not detected)? Why did they not calculate the compound 11 concentrations? If they did not perform adsorption experiments with AC2 and AC3, why did they include columns for compound 11 in Table 2? Those columns seem meaningless.

- The SI file also contains some shortcomings.

The missing NMR spectra of compounds 1-10 are missing.

Figures S1 and S2 do not contain the statistical parameters (equation, correlation coefficient). From Figure S1, a non-linear fit might be more accurate. Please compare the correlation coefficients of linear and quadratic fits.

The structures of Figure S4 are very noisy. Please insert the compound 11 structure in a higher resolution.

The carbon NMR of compound 11 is missing.

The caption of Figure S4 contains a typo.

After including the missing NMR spectra, the authors have to create also a Table of Contents.

Reviewer 2 Report

Authors have made good effort in synthesizing a material with antibacterial action which could have applications in the wound healing. the authors need to address some concerns before the manuscript can be accepted:

1. Introduction needs to be more compact and precise

2. Culture conditions for the bacteria should be provided in method section

3. why did the authors use only one bacteria. wounds and other infection have many bacteria residing at the site. therefore, i recommend to test the material against some more bacteria like E. coli, P.aeruginosa

4. SEM image is not very good. should be more clear in terms of cell morphology

5. Unit for concentration used should be uniform throughout the manuscript either use uM or ug/mL.

6. figure 4 is very confusing. authors need to simplify the illustration

7. table 3 values should be presented as log CFU/mL

Reviewer 3 Report

Dear Editor,

This study, Authors point out the adsorption level of the antimicrobial substance on three different active carbons of different origins; examining the inhibition level of the growth of S. aureus bacteria. It was observed that activated carbons with adsorbed antibiotics show better bactericidal properties than activated carbon alone or the antibiotic itself. In addition, they also synthesized a new antibacterial SufA protease inhibitor.

Follow of the arguments is quite well however there are some points should be taken into consideration before publication.

Comment 1. Graphical abstract. Line 104 Author claim that general inhibition mechanism of the 1-AAP showed as a graphical abstract. However, Graphical abstract should include/indicate the activated carbons and antibiotics insertion possesing dual effect as well. But I should mention here that I could not see the graphical abstract seperately, I gave my comments according to the Authors claim at Line 104.

Comment 2. Carbon activated methods. Notation for the AC1-2, AC1-3 are confusing. Please mention as AC1, AC2 and AC3, otherwise AC1-2 can be understood as carbons were derived from beech wood obtained with different methods. That means for the synthesis of AC1, three methods are applied:AC1-1;AC1-2 and AC1-3. These confusing notations should be revised in whole manuscript accordingly.

Comment 2. Authors mentioned at Introduction part that they previously indicated that the meso- and macropores of the active carbons do not seem to affect the bacterial adsorption process. Is there a similar observation in this study? Is there a relationship between AC-1, AC2 and AC-3 pore size/volume and bacteria adsorption? Please mention at Discussion part.

Comment 3. Line 157, Part 2.4. How is the solution is removed?centrifuge or drying under vacuum?

Comment 4. Line 457, Table 2. Concentration in solution of 11 is “Zero” for AC1 and “ND” for AC2 and AC3. What is the difference between “Zero” and “ND” notation according to the analysis? I think they should be “zero” or “ND” for all sample. Beside that, the instrumental techniques for the concentration in solution should also be cited under the Table 2.

Comment 5. Line 554. Authors claim that the higher content of oxygen groups, the better the sorption of the S.aureus. Is there any prove for the higher oxygen content such as XPS analysis?

Comment 6. Authors gave a conclusion on the presented experiment and its results may be the basis for the introduction of activated carbon with adsorbed antibiotics to plasters, bandages, and sterile dressings in general. However, discussion or future analysis on the cell cytotoxicity for activated carbons functionalized with an antibacterial agent can be mentioned.

Comment 7. Figure S4 1H NMR(Supplemental Information). 1H NMR spectrum peak assignment is missing accroding to the Chem draw of the molecules and residual solvent peaks are present. NMR analysis is recommended to be repeated. Resolution of the insert chemical picture is not good as well.

Comment 8. It is recommended to fix some typos such as MgSO4, 1H NMR, 31P NMR etc.

Comment 9. Title of Part 3.7 should be revised.

Round 2

Reviewer 1 Report

The authors' feedback on the referee's concern is correct, and the modified manuscript is suitable for publication.